

# Advancing medicinal plant agriculture: integrating technology and precision agriculture for sustainability

Vinay Kumar[1,*], Ashwini Zadokar[1,*], Pankaj Kumar[1], Rohit Sharma[2], Rajnish Sharma[1], Mohammed Wasim Siddiqui[3], Mohammad Irfan[4] and Rahul Chandora[5]

[1] Department of Biotechnology, Dr. YS Parmar University of Horticulture and Forestry, Solan, Himachal Pradesh, India
[2] Department of Forest Product, Dr. YS Parmar University of Horticulture and Forestry, Solan, Himachal Pradesh, India
[3] Department of Food Science and Post-Harvest Technology, Bihar Agricultural University, Sabour, Bihar, India
[4] Plant Biology Section, School of Integrative Plant Science, Cornell University, Ithaca, NY, United States of America
[5] ICAR-NBPGR National Bureau of Plant Genetic Resources, Shimla, Himachal Pradesh, India
[*] These authors contributed equally to this work.

Corresponding authors
Pankaj Kumar,
pksharmabiotech@gmail.com
Mohammad Irfan,
mi239@cornell.edu

## ABSTRACT

To strengthen the agriculture sector, it is crucial to combine the efforts of industrialization (field mechanization and fertilizer production), technology (genome editing and manipulation), and the information sector (for the application of current technologies in precision agriculture). The challenge of modern sustainable agriculture is increasing agricultural output while using the least amount of resources and capital expenditure possible and considering the variables contributing to environmental damage. Different environmental factors adversely affect medicinal plant populations, leading to the extinction of these valuable medicinal species. These difficulties drew the attention of the international scientific community to farm sustainability and energy efficiency studies that put forth the idea of precision agriculture (site-specific crop management) in medicinal plants. It is a systems-based method that monitors and responds to changes in intra- and inter-field conditions for environmentally friendly and optimum crop output. Farming systems have significantly benefited from the visualization and morphological analysis of agricultural areas (both open fields and greenhouse experiments) using remote sensing technology, geographic information systems (GIS), crop scouting, variable rate technology (VRT), and Global Positioning System (GPS). These technologies form the backbone of the fourth agricultural technological revolution, Agriculture 4.0. This review concisely summarizes these innovative technologies' current use and potential future advancements in medicinal plants. The review is intended for researchers, professionals in medicinal plant cultivation, herbal medicine research, crop science, and related fields.

## INTRODUCTION

Over the last few decades, manipulating and deciphering the plant genome has advanced significantly more than measuring how plant genetic alterations affect their properties (*Kumar et al., 2023*; *Kaur, Ahmed & Alok, 2024*). The community of scientists that study plants have recently put much work into correcting this imbalance using various molecular, computational, biophysical and interdisciplinary approaches (*Dale et al., 2021*; *Irfan & Datta, 2017*; *Sharma et al., 2023*). For instance, for plant phenotyping several imaging techniques can be used to characterize how plants react to their surroundings (*Sarić et al., 2022*). Plant phenotyping has indeed been identified as a barrier to comprehending the interaction between plants and the environment, managing agricultural systems, and increasing the effectiveness of breeding programs (*Jiang & Li, 2020*). Numerous technological developments let the plant phenotyping process get over these constraints, which generated large datasets that are the source of the plant response (*Asif et al., 2023*). Precision agriculture is at the forefront of shaping itself to offer solutions to the alarming challenges in agriculture (*Alyafei et al., 2022a*). It is the "application of integrated technologies (such as improved machinery, sensors, information systems, and informed management) to optimize productivity by compensating for dynamics under sustainable agricultural systems" (*Yin et al., 2021*). Precision agriculture leverages diverse technological advancements, such as sensing, automation, and data science techniques, to optimize the input of irrigation and chemical applications and enhance the productivity and quality output for an agricultural production system (*Jiang & Li, 2020*; *Alyafei et al., 2022b*). Due to the growing global population, addressing issues like climate change and implementing fast and efficient production is essential. The potential to conserve water and energy, as well as the ability to preserve the environment from pollutants, should all be considered in this process. Precision agriculture accomplishes these objectives by utilizing information and communication technology (ICTs) in agriculture (*Aslan et al., 2022*). Due to the complex needs of our planet, farming and plant breeding are undergoing a rapid transformation. A revolution in agriculture has been sparked by the expansion of collected data to the point where innovation is required (*Id et al., 2022*).

Fast analytical techniques that can simultaneously record multiple chemical and physical data points are needed to guarantee quality from the start, starting with breeding programs on the field and proceeding through the extraction and production processes. Near-infrared (NIR) spectroscopy sensors are advantageous (*Huck, Bec & Grabska, 2022*). Imaging-based plant phenotyping has received greater attention in the last five years due to imaging technologies' enormous promise for high-throughput plant phenotyping (*Jiang & Li, 2020*). High-throughput phenotyping (HTP) (which includes many aerial and ground systems) can provide the highest-quality trait data. These technologies might also be beneficial in precision agriculture by unraveling quantitative traits. Tower-based systems, gantry-based systems, mobile ground systems, low- and high-altitude aerial systems, and satellite-based systems are just a few of the HTP solutions created during the past five years to improve phenotyping capabilities and throughput significantly (*Jiang & Li, 2020*). Different technologies, such as geographic information systems (GIS), global positioning

systems (GPS), and remote sensing, can also be applied in medicinal plant production. GIS has been combined with other approaches to comprehend crop identification, crop area estimation, pest infestation, soil conditions, prospective yield, and agricultural water management (*Gebeyehu, 2019*). These new methods and technologies (GIS, GPS, remote sensing, *etc.*) can only be used successfully if they support ecological protection and sustainable manufacturing. It is possible to discover new techniques for using remote sensing in ecological research from the study of therapeutic herbs (*Neményi, 2015*).

Using new technologies can help farmers optimize input allocation, thereby contributing to lower costs, increased outputs, and higher resource efficiency. More precisely, the use of sensors can contribute to better monitoring of a farm so that inputs, such as fertilizers or pesticides, can be applied according to its needs (*Walter et al., 2017*), assuming that the farmer can use the data collected on the farm and put it into practice. Nevertheless, adoption rates for precision agriculture technologies (PAT) vary significantly among geographical locations and various technologies, despite this technology's potential (*Barnes et al., 2019*; *Lowenberg-DeBoer & Erickson, 2019*). The medicinal plant market is diverse, encompassing raw plant materials, herbal extracts, essential oils, and processed formulations used in pharmaceuticals, cosmetics, and dietary supplements. With the global demand for herbal medicines surging, approximately 3,000 species are actively traded internationally (*Rajeswara Rao & Rajput, 2010*; *Mudge, Betz & Brown, 2016*). As per recent market research future analysis, the global herbal medicine market, valued at USD 100.12 billion in 2024, is expected to grow to USD 349.61 billion by 2034, with a compound annual growth rate (CAGR) of 13.32% from 2025 to 2034. This growth is driven by the rising consumer awareness of the adverse effects of allopathic medications and the increasing prevalence of chronic diseases (*Market Research Future, 2024*). However, the sustainability of medicinal plant populations remains a challenge due to overharvesting and habitat destruction (*Rajeswara Rao & Rajput, 2010*). The populations of medicinal plants are being negatively impacted by habitat loss, excessive harvesting, global warming, and climate change, which threaten the future existence of 15,000 species. Cultivation is becoming a financially viable solution to preserve medicinal plants' diversity and global abundance (*Rajeswara Rao & Rajput, 2010*). High-quality herbal goods are still in demand (*Albadwawi et al., 2022*), and according to estimates from the American Herbal Products Association (AHPA), 3,000 plant species are currently marketed (*Mudge, Betz & Brown, 2016*). Modern precision approaches, which enable the effective *in-situ* and *ex-situ* conservation of medicinal plants, are faster and more efficient than traditional morphological and physiological data processing and analysis (Fig. 1). This review provides insights on cutting-edge technologies utilized in the cultivation and conservation of medicinal plants.

## SURVEY METHODOLOGY

We used various academic articles search engines including PubMed, Google Scholar, Web of Science, Science Direct, and Scopus databases. For literature survey, the key terms such as "Role of GPS and GIS in Medicinal plants, Precision agriculture, Sensor technology, Variable rate technology" were used. Based on this search criteria, we retrieved 69 publications sourced from the above search engines. Only articles written in English
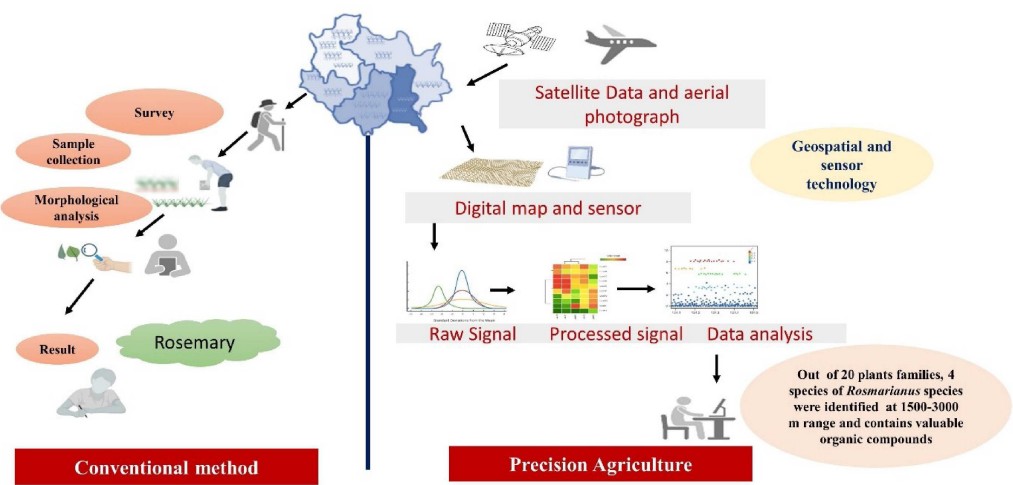

**Figure 1** Overview of conventional method *versus* precision agriculture in the medicinal plant (**Rosemary as a case study**).

were considered. The abstract of all these articles were analyzed for the suitability for inclusion in the article. This captured a wide yet relevant dataset on the integration of geospatial technologies and sensor innovations in medicinal plants research.

For the case studies, we analyzed 15 core research studies published over the period 2005 to till date; these collectively represent the developments in remote sensing and sensor technologies applied in this area. Our survey indeed suggests that the relevant publication has increased gradually with time, with the early studies appearing rather sporadically until an obvious increase in output between 2017 and 2019. Indeed, 2019 stands out as a singularly remarkable year, as it bears witness to four major studies that contribute to the field.

This analysis provides insight into the progressive adoption of advanced technologies that largely underpin improvements in access, distribution, and conservation at the cultivation and utilization ends of the medicinal plants' value chain. The pattern that is beginning to emerge here is the increasing awareness of the potential of GPS, GIS, and sensor technologies in fostering sustainable practices and methods of precision agriculture in this field, hence improving its potential for meeting current and future demands for its products.

## GEOSPATIAL TECHNOLOGIES

Geospatial technology is used to acquire information positioned on the earth for monitoring, tracing, analysis, modeling, simulations, and visualization. It is a satellite-based geolocation system that correlates an object's location with its coordinates. It provides informed decisions based on the importance and priority of resources that are limited in nature. Basic geospatial technology includes GPS, GIS, and remote sensing. In agriculture, geospatial technology could be used in eco-geographic surveys, field exploration, site

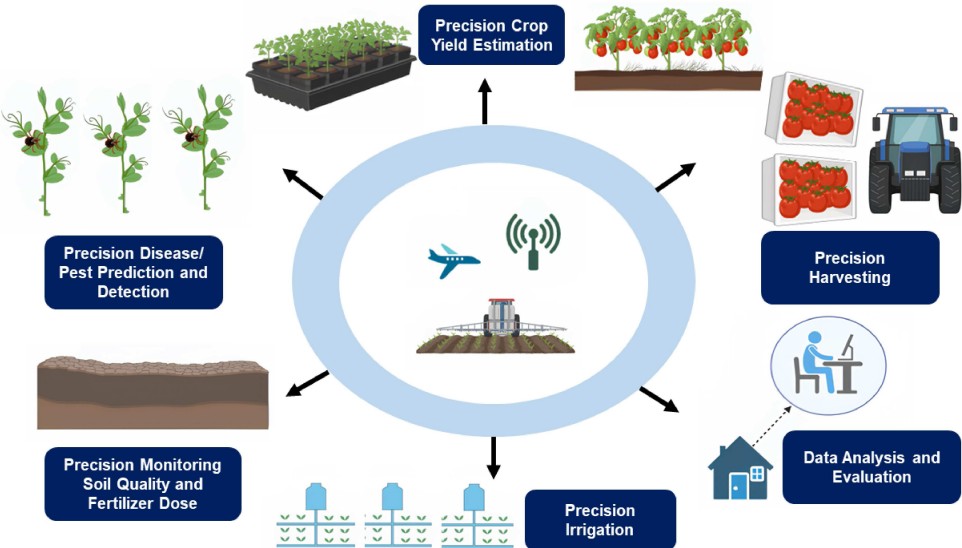

**Figure 2** Application of geospatial technology in agriculture.

identification, germplasm conservation, and the distribution of threatened species' genetic resources in nature (*Gixhari et al., 2012*; *Chong et al., 2017*). Precision agriculture leverages diverse technological advancements, such as remote sensing, automation, and data science techniques (GPS and GIS), to optimize the input of irrigation, harvesting, soil quality monitoring, pest prediction, crop yield estimation, and chemical applications, and to enhance the productivity and quality output of an agricultural production system. In the future, precision farming will harness a range of advanced technologies, including remote sensing, automation, and data science tools like GPS and GIS, to revolutionize the cultivation of medicinal plants. These innovations will enable the optimization of critical processes such as irrigation management, harvesting, soil quality monitoring, pest prediction, crop yield estimation, and the application of chemicals. By integrating these cutting-edge techniques, precision farming will enhance both the productivity and quality of outputs in agricultural production systems (Fig. 2).

Currently, overexploitation, inequitable uprooting, habitat destruction, population declines, and a lack of proper management practices threaten medicinal plants and their diversity. Spatial tools, which can provide crucial information on diversity present in specific geographic areas that can be used for analysis and map generation, facilitate the effective outcomes for gene bank management and encourage the development and execution of conservation policies (*Guarino et al., 2002*). In the past, several researchers used geospatial tools like remote sensing, geographic information system, and global positioning system used to acquire data for surveying, identifying, classifying, mapping, monitoring, analyzing, modeling, simulating, characterizing, tracking, and managing several threatened species and their distribution (Table 1) (*Moraes et al., 2005*; *Mashayekhan et al., 2016*; *Saadi et al., 2017*; *Biswas, Walker & Varun, 2017*; *Qose & Proko, 2018*; *Cerasoli et al., 2018*; *Tshabalala et al., 2021*). In India, NMPB and the Ministry of AYUSH have started two projects under

Kumar et al. (2025), *PeerJ*, DOI 10.7717/peerj.19058

**Table 1  Application of geospatial technologies (GIS & GPS) in different medicinal plants.**

| Database system | Developer | Study area | Target species | Results | References |
|---|---|---|---|---|---|
| DGPS | USCG & CCG | Mississippi, USA | *Podophyllum peltatum* | Selection of elite genotypes and evaluation of biotic & abiotic factor that effect drug yields | *Moraes et al. (2005)* |
| ArcGIS 9.3 | ESRI, USA | Darkesh, Iran | 10 medicinal plant species | Conservation and consideration of endangered medicinal plants of Darkesh forest | *Mashayekhan et al. (2016)* |
| GPS | US | Vidyasagar University campus | 156 medicinal plant species | Determined the diversity of medicinal plants | *Saadi et al. (2017)* |
| ArcGIS 10.2 | ESRI, USA | St. John's College, Agra, India | 33 families of medicinal plants | Enhance conservation strategies | *Biswas, Walker & Varun (2017)* |
| GMPGIS | Institute of Chinese Materia Medica, China Academy of Chinese Medical Sciences | U.S., Global Biodiversity Information Facility, Kew Gardens National Specimen Information Infrastructure & Chinese Virtual Herbarium | *Panax quinquefolius* | Introduction, conservation and cultivation in new ecologically suitable area | *Liang et al. (2019)* |
| GPS & GIS | US | USU Arboretum, Pancur Batu Sub District, Deli Serdang District, North Sumatra Province, Indonesia | 17 species | Enhance diversification | *Rahmawaty et al. (2019)* |
| ArcGIS | ESRI, USA | Bahar Mountain catchment | *Astragalus species* | Determine factors that can used for identification of cultivation area | *Piri et al. (2019)* |
| GMPGIS | Institute of Chinese Materia Medica, China Academy of Chinese Medical Sciences | China, Royal Botanic Gardens & Global Biodiversity Information Facility | *Crocus sativus* | Enhance globalisation of the Chinese herbal medicine planting industry | *Wu et al. (2019)* |

Kumar et al. (2025), *PeerJ*, DOI 10.7717/peerj.19058

**Table 1** (*continued*)

| Database system | Developer | Study area | Target species | Results | References |
|---|---|---|---|---|---|
| GIS | US | Pushparajgarh, Madhya Pradesh, India | *Mapping of antimalarial plants* | Nineteen antimalarial plants from 13 families and 19 genera have been identified | *Dwivedi et al. (2020)* |
| GPS and GIS | US | Five villages of the Great Zab in Province in Kurdistan Region, Iraq | distribution of medicinal plants | 21 families and 44 individual species of plant were found. The largest number was in the Liliaceae family, accounting for seven species, followed by Lamiaceae and Asteraceae with four species in each family | *Faizy et al. (2023)* |
| QGIS | International team of developers (from USA, Switzerland and middle Europe) | Karnataka, India (latitudes 11°30′N to 18°30′N and longitudes 74°E to 78°30′E) | *Aphanamixis polystachya Cinnamomum macrocarpum Plectranthus caninus Jatropha gossypiifolia Zinnia peruviana Exacum bicolor Carica papaya* | Developed a comprehensive dataset of over 160,000 entries, facilitating targeted recommendations for cultivating medicinal herbs based on soil types and subregions | *Roopashree et al. (2024)* |

**Notes.**

GMPGIS, Global Medicinal Plant Geographic Information System; ArcGIS- Arc, Geographic Information System; ESRI, Environmental Systems Research Institute; DGPS, Differential Global Positioning System; USCG, United States Coast Guard; CCG, Canada Coast Guard; U.S. & USA, United State of America; USU, Universitas Sumatera Utara; GPS, Global Positioning System; GIS, Geographic Information System; QGIS, Quantum GIS.

its central sector scheme for the conservation of medicinal Plants and their management, *i.e.,* "Geospatial approach for suitable site identification for conservation of some species of Medicinal and Aromatic Plants (MAPs) at selected districts of Uttarakhand" at Uttrakhand Space Application Centre, Dehradun and another one's "Inventorization, Digitization and web enabling of the Geospatial Maps of Medicinal and Aromatic Plants cultivated in the States of Andhra Pradesh, Tamil Nadu, Karnataka, Kerala, and Orissa" at Central Institute of Medicinal and Aromatic Plants (CIMAP), Lucknow.

## Global positioning system

A radio navigation system using satellites and computers to find out the altitude, latitude, and longitude of an object on the earth by analyzing their time difference based on the reception of signals from an array of satellites (24) to the object. It is capable of providing a location accuracy of one m. The US government started the Navstar (global positioning system; GPS) technology for defense use in 1973, but later in 1983, it was extended for civilian use. During 2013–2017, the Indian Regional Navigation Satellite System (IRNSS) launched seven satellites into space with the operational name NAVIC (Navigation with Indian Constellation), providing precise real-time positioning and timing. In 2018, the Indian Space Research Organization (ISRO) operated NAVIC at two levels, *i.e.,* one for civilian use (standard positioning service) and the other for authorized users (restricted service). Service called GLONASS is operated by Russia and its satellites orbits on three different planes. Each orbit has eight satellites used by the military, and civilians also use some open signals. Another GNSS is Galileo, developed in 2019 for civilian use and has three orbits with ten satellites in each orbit. The Chinese GNSS is known as the BeiDou Navigation Satellite System (BDS) with six orbital planes. It has more satellites than GPS or any other system. In agriculture, numerous GPS-based tools help farmers make their fields more productive and efficient for precision farming. GPS-derived products are inexpensive, easy to use, and accurate, allowing farmers to develop maps with particular areas, locations, and distances between points of interest. It also allows farmers to navigate and monitor that location for soil sampling, irrigation, fertilizer utilization, pest, weed, and disease management, and minimizing chemical drift. In the recent year, several GPS tools have been used in many forest areas for survey, positioning, and identification of specific locations & habitats of plant species that were classified as medicinal plants or protected species (*Mashayekhan et al., 2016*; *Saadi et al., 2017*; *Biswas, Walker & Varun, 2017*; *Liang et al., 2019*; *Rahmawaty et al., 2019*; *Piri et al., 2019*; *Wu et al., 2019*).

In Darkesh (Iran), GPS was used for field verification to map medicinal plants. The data was integrated into ArcGIS 10.8. The data represented approximately 1,100 medicinal plants of Iran, of which 140 plant species in Darkesh and only ten medicinal plant species were used for trading in the market. Trading showed a 10% increment in household income, reducing economic vulnerability and increasing livelihood and employment options (*Mashayekhan et al., 2016*). Also, a portable GPS unit was tested against traditional field methods at Vidyasagar University Campus, Midnapore, West Bengal, India. Results from the GPS accuracy test showed that the most significant average deviation from the proper position for the surrounding area was 48 m from the mean sea level and indicated

that the operator with a minor level of plot-location expertise was nevertheless able to locate plots more quickly through GPS than by using traditional methods. As reasonable impediments, it lists system portability, satellite accessibility, canopy signal interference, and operator biases. In the South West Bengal region of Paschim Medinipur, India, their study recorded a variety of medicinal plants on the campus of Vidyasagar University and in the surrounding area. This comprises 108 genera, 117 species, and 49 plant families, divided into the Marchantiophyta, Pteridophyta, Monocotyledones, and Dicotyledones groups. Therefore, GPS technology was used to conserve medicinal plant species from anthropogenic activities (*Saadi et al., 2017*).

## Geographic information system

It is a database for spatial information management containing geographical data combined with different software tools for monitoring, managing, analyzing, and mapping. Users claim that geographic information system (GIS) is a multi-level digital mapping system in which input data can come from satellite, airborne, or UAV-based image collection sources. Different mathematical tools are applied for processing and analyzing data at different layers. Therefore, GIS is a database system that allows us to monitor, question, interpret, and visualize data through maps, globes, reports, patterns, and charts (*Fadeev et al., 2019*).

GIS technology analyzes large volumes of spatial data from sources like field surveys, evaluation, documentation, remote sensing, and already developed maps and reports on medicinal plants' accessibility, distribution, and conservation. Data input, encoding, management, analysis, manipulation, and presentation are the major GIS components that combine information from various sources into a single platform. Mostly, GIS can be useful for *in situ* conservation by designing on-farm conservation sites, utilizing existing *ex-situ* collections, identifying gaps in *ex-situ* collections, and developing core sets and propagation techniques for various species in *ex-situ* collections (*Moraes et al., 2005*; *Mashayekhan et al., 2016*; *Saadi et al., 2017*). It can also represent distribution maps of threatened medicinal plants by assessing *ex-situ* collections in the genebank database (*Liang et al., 2019*; *Wu et al., 2019*). GIS technology helps researchers identify pest pathogen diversity, imitate early warning systems, build risk assessment models, discover hotspots and medicinal plant conservation areas free from pest attacks, *etc.* (*Guarino et al., 2002*; *Moraes et al., 2005*; *Piri et al., 2019*). Previously, several researchers used the GIS database system for mapping *in situ* and an *ex-situ* collection of various medicinal plants (Table 1).

The variety and abundance of medicinal plant species found nationwide result from climatic variance and various ecological situations. Based on the region's characteristics, field surveys, information from the area, and expert opinions were the most valuable criteria for identifying and distributing medicinal plants. Primary criteria include precipitation, temperature, slope, altitude, soil texture, and orientation. In the medicinal plant *Astralagus hypogean* Bunge, GIS, and analytic hierarchy process (AHP) technologies were used to discover their locations in the Bahar Mountain catchment with the least effort and expense (*Piri et al., 2019*). GIS technology was also used to determine the diversity and distribution of medicinal plants. In the Universitas Sumatera Utara (USU) botanical garden, the GIS

system was used to map the distribution of medicinal plants in the arboretum and learn about the many plant components used in medicine. The techniques utilized to gather the data were surveys & interviews, and the Shannon-Wiener index was used to calculate the species diversity (*Keylock, 2005*). The findings revealed that the arboretum had 12 poles, 21 types of trees, and 17 species of medicinal plants. The state of the arboretum suggests there was still room for more medicinal plants to be planted in order to boost species variety, particularly in the western and northern sections (*Rahmawaty et al., 2019*).

Since ancient times, Indian culture has included the usage of medicinal herbs and knowledge of their properties. They can grow in harsh climates and are found naturally. However, their excessive exploitation and forced removal showed concern about their issues with identification and mapping. St. John's College of Agra was chosen for the GPS investigation to recognize and map the precise GPS position of the local medicinal plant species. Investigation showed that 56 plant species from 33 families had been identified, and their precise GPS coordinates had been recorded to map their global distribution using a variety of themed maps created in a GIS context by ArcGIS 10.2. After combining the attribute data with the thematic maps, ArcGIS Online released the dynamic interactive Web application for public usage on the Web GIS platform (*Biswas, Walker & Varun, 2017*). Based on their 476 occurrence sites and 19 bioclimatic factors, the Geographic Information System for Global Medicinal Plants (GMPGIS) was also used to estimate the locations and distribution of Panax quinquefolius and how well it might fit in with the environment. GIS mapping analysis showed that by 2070, the environmentally suitable planting areas for *P. quinquefolius* expanded north and west relative to the existing ecologically suitable regions with the addition of 9.16% to 30.97%, and also showed that the Lesser Khingan Mountains of China, northern Canada, and eastern Europe were the prospective enhanced zones that were environmentally suited as compared to central China, the southern United States, and southern Europe. Jackknife experiments show that precipitation during the hottest quarter was the main climatic factor influencing *P. quinquefolius* distribution (*Liang et al., 2019*). Previously, several researchers used the GIS database system for mapping *in situ* and an *ex-situ* collection of various medicinal plants (Table 1).

## Remote sensing

It is a technology in which the physical properties of an object, area, or any phenomenon can be detected, monitored, identified, and analyzed by measuring its spectral reflectance and thermal emittance at a distance without forming direct contact with them. The leading platforms in remote sensing are aircraft and satellites (*Sabins, 1999*). Land use mapping, weather forecasting, crop management, environmental studies, disease management, natural disaster studies, and resource exploration are the primary applications of remote sensing techniques (*Ray, 2016*). Two approaches, *i.e.,* multispectral and hyperspectral imaging, can be used in remote sensing. The multispectral imaging system can differentiate visible ranges into 3–6 bands. It can provide information about vegetation, soil types, irrigation, and other selected farmer objects, while hyperspectral imaging divides the electromagnetic spectrum into hundreds or thousands of narrow bands (10–20 nm). The information from hyperspectral imaging is more precise and addresses all the

farmer's issues (*Shippert, 2003*; *Shippert, 2004*; *Govender et al., 2008*; *Govender et al., 2009*). High-resolution satellite imagery, multispectral channels, and aerial photography (UAV-uncrewed aerial vehicles) increase the research on several medicinal plants and their resources for larger scales (Table 2) (*Qose & Proko, 2018*; *Cerasoli et al., 2018*; *Tshabalala et al., 2021*; *Othman et al., 2015*).

An empirical modeling technique was used in Mediterranean grasslands to estimate the gross primary production (GPP) receiving various fertilization treatments from hyperspectral reflectance. The study aimed to determine which combinations of vegetation indicators and bands in Mediterranean grasslands depict GPP fluctuations between the annual growth peak and senescence dry-out. Similar explanatory capabilities for the two simulated satellite sensors were found when vegetation indices and bands were included in the models, compared between the Sentinel-2 and Landsat 8-based models. In order to promote sustainable agricultural management, remote sensing techniques could follow the phenology of plants and improve GPP estimates (*Cerasoli et al., 2018*). The use of modern geo-IT tools and remote sensing has become increasingly valuable for inventorying, sustainably managing, and monitoring medicinal and aromatic plants on the municipal grounds of Skrapari, situated in Albania's southeast. Six hundred forty-seven vascular plants had been identified and inventoried, with 111 families, a wide range of variation in their chronological and biological forms, endemic and sub-endemic species from the Balkans, and the presence of species with endangered status (IUCN) (*Qose & Proko, 2018*).

In precision agriculture, remote sensing data provide a powerful tool to measure variability within the field, evaluate crop health, irrigation schedule, stress, nutrients, pest management, weed monitoring, yield prediction, optimize the use of agrochemicals, and also provide information on the surrounding area in the investigating field (Table 2). Remote sensing hyperspectral data may be used to forecast the biomass yield of several *Moringa oleifera* cultivars, providing insight into forecasting medicinal plants' biomass output utilizing spaceborne and airborne data by extrapolating hyperspectral data to wider sizes (*Tshabalala et al., 2021*). The remote sensing technique was also used for a resource survey of the cultivated medicinal plant *Panax notoginseng*. A set of survey methods for *P. notogingseng* based on this technique was developed, including the remotely sensed data-source selection, remote sensing image processing, interpretation, and validation (*Zhou et al., 2005*). Remote sensing techniques may also identify and scale physiological responses at the leaf level to cover vast regions, providing essential and trustworthy data on water usage and irrigation management. Similarly, in two southern New Mexico pecan orchards and *Carya illinoinensis*, a remote sensing technique was used to measure the moisture status by plant physiological responses, and remotely sensed surface reflectance data were collected from either well-watered or water-deficient trees (*Othman et al., 2014*; *Othman et al., 2015*). In *Valeriana jatamansi*, hyperspectral remote sensing data with analytical spectral devices (ASD) was used for age discrimination and offers a methodical approach for harvesting this plant at its prime age while minimizing wastage. Therefore, the remote sensing technique was used to determine the best growing stages for harvesting precious and endangered medicinal plants (*Kandpal et al., 2021*).

Kumar et al. (2025), *PeerJ*, DOI 10.7717/peerj.19058

**Table 2** Application of remote sensing technology in various medicinal plants.

| Platform | Target species | Study area | Data analysis method | Output | References |
|---|---|---|---|---|---|
| SENTINEL-2 | Medicinal plant | South-East Albania | Juice Program 7.0 & TURBOVEG program | Invention of endangered medicinal plants | Qose & Proko (2018) |
| SENTINEL-2 | Mediterranean grasslands | Companhia das Lezírias, north-east of Lisbon, Portugal | Open-source R & MLR models | Enhance productivity and leaf area | Cerasoli et al. (2018) |
| FieldSpec spectroradiometer | *Moringa oleifera* | Agricultural Research Council experimental farm in Roodeplaat, Pretoria, South Africa | ANOVA, RRMSEV, RF classification ®ression | Maximum prediction of biomass yield | Tshabalala et al. (2021) |
| SPOT-5 | *Panax notogingseng* | Maguan county, China | – | Determine relative right rate of judgement, the relative precision of area and the relative precision of yield | Zhou et al. (2005) |
| Landsat 8 | *Carya illinoinensis* | Mesilla Valley near Las Cruces, New Mexico | Pearson's correlation and MLR using SAS version 9.3 software | Determine leaf area, chlorophyll content, water stress, photosynthesis rate and leaf gas exchange | Othman et al. (2014) |
| ASD handheld Spectroradiometer | *Valeriana jatamansi* | IHBT, Palampur, India | PCA, RF classification, BDT, Decision Tree DT and Knn | Age determination and discrimination | Kandpal et al. (2021) |
| Fieldspec Pro Full Range Spectroradiometer | *Carya illinoinensis* | Mesilla Valley near Las Cruces | Boxplot analysis & Discriminant analysis using PROC DISCRIM in SAS version | Determine leaf area, chlorophyll content, water stress, photosynthesis rate | Othman et al. (2015) |
| Mask R-CNN on ultra-high resolution UAV and DL | *Lamiophlomis rotata* | Qinghai-Tibet Plateau | Linear regression analysis, ResNet-101 and ResNet-50 | Developed an LR dataset from UAV images and proposed a new pipeline for wild medicinal plant detection and yield assessment based on UAV and DL | Ding et al. (2023) |

**Notes.**

ANOVA, Analysis of variance; RF classification, Random Forest classification Relative Root Mean Square Error of Validation; MLR, Multiple Linear Regression; IHBT, Institute of Himalayan Bioresource and Technology; DT, Decision Tree; BDT, Boosting Decision Tree; Knn, k-Nearest Neighbourhood; UAV, Uncrewed Aerial Vehicle; DL, Deep Learning; R-CNN, Region Convolutional Neural Networks; LR, *Lamiophlomis rotata*.

## Proximal sensors

A sensor is a device that responds to a physical stimulus from its environment. That physical stimulus could be heat, light, sound, pressure, or other environmental phenomena and transmits it into a signal easily interpreted by others (humans and machines) for further processing. Sensors are of two types, *i.e.,* analog sensors, which convert physical data into an analog signal, and digital sensors, which work with digital data. Analog sensors are more precise than digital ones because digital sensors are limited to a finite set of possible values (0, 1) while analog is continuous (*Henry, 2000*). Different data analysis methods were used to interpret and process the complex data sets from the signals. These are of two types, *i.e.,* linear response methods like hierarchical cluster analysis (HCA), principal component analyses (PCA), soft independent modeling of class analogy (SIMCA), linear discriminant analysis (LDA), partial least squares (PLS), cluster analysis (CA), discriminant functions analysis (DFA) and canonical discriminant analysis (CDA) (*Gardner, 1991*; *Goodner, Dreher & Rouseff, 2001*) & other non-linear response methods that can be widely used for classification and regression. Many pattern recognition techniques were used for non-linear responses, like the radial basis function (RBF), k-nearest neighbor (k-NN), support vector machine (SVM), artificial neural network (ANN), self-organizing map (SOM), and relevance vector machine (RVM) (*Distante, Ancona & Siciliano, 2003*; *Wang, Zhang & Zhang, 2005*).

In agriculture, sensor devices are developed to measure soil properties, stress, yield, growth condition, irrigation, weed, pest, and disease infestation by sensing the field's external stimuli (*Seregély & Novák, 2005*; *Govender et al., 2009*; *Islam et al., 2006*). Medicinal plants play an essential role in human health and society. The quality and quantity of medicinal plants correlate with the remedies used with them (*Xylia et al., 2022*). Therefore, several sensor systems were used to maintain the quality and quantity of medicinal and aromatic plants for, *e.g.*, pH sensors that can monitor soil pH and nutrient levels (*Carmona et al., 2006*; *Islam, Ahmad & Shakaff, 2010*; *Banal et al., 2014*; *Xiong et al., 2014*). Moisture sensors can control irrigation systems and water levels (*Zou et al., 2014*; *Ray et al., 2014*; *Sytar et al., 2015*). Electronic sensors can detect odors; light sensors can adjust illumination & misting frequency; infrared sensors can determine leaf temperature to predict moisture stress; and leaf wetness sensors can predict fungal spore growth and disease severity, *etc.* (*Taha & Abu-Khalaf, 2020*; *Okur et al., 2021*; *Patle et al., 2022*) (Table 3).

Sensor technology has greater importance for medicinal and aromatic plants to improve their quality and quantity parameters (*Oh et al., 2008*; *Kataoka et al., 2008*; *Oh, 2013*; *Cui et al., 2015*; *Peng et al., 2014*; *Zheng, Ren & Huang, 2015*; *Tsuchitani et al., 2022*). Sensory characteristics like odor have marketable importance in spices and crude drugs from medicinal plants Therefore, instrumental sensory system analysis has proved to be a novel and efficient method in chemometrics. An electronic nose called an "e-nose" sensor system can detect odors and flavors. For example, the NST-3320, AppliedSensor Sweden AB E-nose system was used to distinguish the aroma of essential oils extracted from samples of oregano and lovage (*Seregély & Novák, 2005*) *Jasminum sambac, J. auriculatum & J. grandiflorum* (*Ray et al., 2014*). It may be a proper technique with the combination of multivariate methods (PCA & CDA) to distinguish between different ages or harvesting

Kumar et al. (2025), *PeerJ*, DOI 10.7717/peerj.19058

**Table 3** Application of sensor technology in medicinal and aromatic plants to improve their quality and quantity parameters.

| Type of sensor | Planting materials | Location | Data analysis | Function | References |
|---|---|---|---|---|---|
| Artificial lipid-polymer membrane sensor | *Eurycoma longifolia* | Forest Research Institute of Malayisa | HCA & PCA | Qualitatively determining the maturity stage, batch-to-batch variation, different parts of plant and mode of extraction | *Ahmad et al. (2006)* |
| Chemosensor (E-nose) | *Origanum vulgare* subsp. *hirtum* and *Levisticum officinale* | Budapest | PCA & CDA | Identification to accelerate selection process & also used to differentiate drugs at different age & harvesting period. | *Seregély & Novák (2005)* |
| SAW sensor | Thymus species | South Korea & Europe | PCA | Detection of volatile compounds | *Oh et al. (2008)* |
| Artifical taste sensor (SA402) | 11 medicinal plants and 10 Chinese medicines | Osaka, Japan | Euclidean Distance | Evaluate berberine content | *Kataoka et al. (2008)* |
| SAW sensor | Lavandula species | England, Australia, France & USA | PCA | Detection of volatile compounds | *Oh (2013)* |
| MOS (E-nose) | *Panax ginseng* | China | DFA & PCA | Detection of volatile compounds at different growth stage | *Cui et al. (2015)* |
| MOS (E-nose) | Zingiberaceae | Beijing, China | PCA | Classification of 10 different species | *Peng et al. (2014)* |
| MOS (E-nose) | *Angelicae sinensis* | China | PCA & DFA | Plant discrimination | *Zheng, Ren & Huang (2015)* |
| Scent sensor FF-2A (E-nose) | 15 medicinal herbs | Japan | HCA | Qualification and quantification of fragrance | *Tsuchitani et al. (2022)* |
| MOS (E-nose) | *Crocus sativus* | Spain, Iran, Greece & Morocco | PCA | Detection of volatile compounds from different region | *Carmona et al. (2006)* |
| QCM array sensor (E-nose, APZ-L, PPG1200 and TOMA s) | *Eurycoma longifolia* | Perlis, Malaysia | PCA | Detection of organic volatiles | *Islam, Ahmad & Shakaff (2010)* |
| E-nose sensor | *Vitex negundo, Mentha arvensis, M. piperita, Artemisia dracunculus Blumea balsafimera, and Plectranthus amboinicus* | Manila, Philippines | PCA & DA | Plant discrimination | *Banal et al. (2014)* |

**Table 3** (*continued*)

| Type of sensor | Planting materials | Location | Data analysis | Function | References |
|---|---|---|---|---|---|
| MOS (E-nose) | *Lonicera japonica* | China | PCA, RBF & LDA | Determine variation of odors in Lonicera stored at different period | *Xiong et al. (2014)* |
| MOS (E-nose) | Asteraceae | Beijing, China | RBF & PCA | Authentication and Classification of different species | *Zou et al. (2014)* |
| MOS (E-nose) | *Jasminum sambac, J. auriculatum, J. grandiflorum* | Tamil Nadu, India | LDA | Characterization of various aromatic compounds | *Ray et al. (2014)* |
| Multiplex fluorimetric sensor (MPx) | Medicinal plants from *Asteraceae, Lamiaceae & Rosaceae* | University in Nitra, Slovakia | FLAV index, Optical fluorescence apparatus Multiplex | Evaluate total flavonoid contents | *Sytar et al. (2015)* |
| ET sensor | "Relax" medicinal product | Bajjora Company, Tulkarm, Palestine | HCA & PCA | Evaluate shelf life | *Taha & Abu-Khalaf (2020)* |
| QCM array sensor (E-nose) | Lamiaceae family (basil, mint & lemongrass) | Germany | LDA | Plant discrimination | *Okur et al. (2021)* |
| Graphene oxide LWS | *Ocimum tenuiflorum* | – | Phytos 31 | For disease supervision | *Patle et al. (2022)* |
| ML-based electrochemical fingerprinting platform | Detection of 6 species of *Anoectochilus roxburghii* | Jinhua Academy of Agricultural Sciences in Zhejiang Province, China | – | Identification and electrochemical fingerprinting | *Xiao et al. (2023)* |
| CMOS+RGB sensor | *Ligusticum chuanxiong* Hort. | Medicinal Botanical Garden of Chengdu University of Traditional Chinese Medicine, China | Python 3.8, LR, NBM, SVM, DT, and RF | Developed a nutrient deficit recognition technology based on UAV multispectral images | *Li et al. (2023)* |
| Ultrasonic sensor (SRF04) | *Hyssopus officinalis L.* | Iran | | Developed a precision harvesting unit | *Saebi et al. (2024)* |

**Notes.**

E-nose, Electronic nose; PCA, Principal Component Analysis; CDA, Canonical Discriminant Analysis; HCA, Hierarchical Cluster Analysis; DFA, Discriminant Functions Analysis; LDA, Linear Discriminant Analysis; RBF, Radial Basis Function; DA, Dendrogram Analysis; ET, Electronic tongue; LWS, Leaf Wetness Sensor; SAW, Surface acoustic wave; MOS, Metal Oxide Sensor; QCM, Quartz Crystal Microbalance; ML, machine learning; CMOS, Complementary Metal Oxide Semiconductor; UAV, Uncrewed Aerial Vehicle; LR, Logistic Regression; NBM, Naive Bayesian Model; SVM, Support Vector Machine; DT, Decision Tree; RF, Random Forest.

times of crude pharmaceuticals from identified cultivars to speed up the selection process. For example, ten distinct Chinese herbal medicine species from the *Zingiberaceae* family were also identified based on their E-nose response signals (*Peng et al., 2014*).

E-nose has been created to detect organic volatiles based on a quartz crystal microbalance array sensor with varieties of polar and non-polar volatile sensitivities (*Islam et al., 2006*). It was an intriguing method of assessing the quality of medicinal plant extract in different medicinal plants like *Eurycoma longifolia,* basil, mint, lemongrass, and various *Lamiaceae* family plants to show its capacity to recognize minute variations in their volatile compounds (*Islam, Ahmad & Shakaff, 2010*; *Okur et al., 2021*). By combining chemometric analysis, GC-MS methods with E-nose were able to study the aroma profiles of different planting materials (*Panax ginseng, Angelica sinensis, Crocus sativus, Malva sylvestris, Matricaria chamomilla, Hibiscus sabdariffa, Mentha piperita, Tilia europaea, Echinacea purpurea, Sambucus nigra, Hypericum perforatum, Taraxacum officinale, Urtica dioica, Passiflora incarnata, Alex paraguayensis, Morus alba, Rubus idaeus*, and *Rosa canina*) at various ages (*Cui et al., 2015*; *Zheng, Ren & Huang, 2015*; *Tsuchitani et al., 2022*; *Carmona et al., 2006*). For the differentiation of medicinal plants, an electronic nose based on chemiresistors and using conducting polymers as the sensing material was created. The conducting polymers were created using potentiostatic electropolymerization and placed on a Teflon substrate between two gold wires 200 m apart. Several negative ions were doped into the polymers. After exposure to the headspace of the finely cut leaves of the following medicinal plants, *i.e., Vitex negundo, Mentha arvensis, M. Piperita, Artemisia dracunculus, Blumea balsafimera,* and *Plectranthus amboinicus*, the resistance of the doped polymers was altered. Therefore, the sensor array responded differently to the various plant samples, and then pattern recognition methods like radar plots, PCA & DA were applied for data analysis (*Banal et al., 2014*).

For non-destructive flavonoids estimation, the optical fluorescence instrument Multiplex(R) 3 (Force-A, France) sensor system was used to detect fluorescence. The FLAV index (expressed in relative units), which is derived from flavonoid's UV absorption capabilities was used to determine the quantity of total flavonoids (*Sytar et al., 2015*). For the purpose of detecting volatile fragrance molecules released from different medicinal plants (*Thymus* (*Oh et al., 2008*) and *Lavandula* family (*Oh, 2013*) fast gas chromatography in combination with a surface acoustic wave sensor (GC/SAW) had been used to separate and characterizes odours-based on subtle differences in their chemical composition due to their botanical and geographic origin using the scent pattern analysis.

Storage time and quality of different medicinal plants were also monitored through an E-nose sensor system. In *Lonicera japonica,* the E-nose approach might be a rapid, simple, precise, and effective technique for quality assessment. Along with the storage length, the odor response and the chlorogenic acid content decreased, showing a strong association between chlorogenic acid concentration and the odor index (*Xiong et al., 2014*). To fully integrate phytomedicine into the primary healthcare system, quality monitoring of herbal medications continues to be a complex problem. A quick and economic evaluation approach to describe the chemical fingerprint of the plant without undergoing time-consuming sample preparation was disclosed, since medicinal plants are complex systems

of combinations. Therefore, new technology based on fabricated multichannel sensors incorporating an arrangement of the synthetic lipid-polymer membrane was used as a fingerprinting device for quality evaluation of different medicinal plants. In *Eurycoma longifolia*, a sensor array system based on the idea of a bioelectronic tongue, which uses synthetic lipid material as a sensing element to replicate the human gustatory system, was used to develop potentiometric fingerprint profiles from various plant sections, age extract, batch-to-batch fluctuation, and other extraction techniques of *E. longifolia* by characterizing the potential electrical response depicted by radar plot. Hence, based on a potentiometric examination of the fingerprints of phytomedicine, the sensor is a viable analytical tool for quality monitoring (*Ahmad et al., 2006*).

Global health and social values depend on the utilization of medicinal plants. The effectiveness of these products depends on their quality. These products' content and shelf life are tracked using various analytical methods. The quality of a locally made herbal remedy called Relax, which contains oats, wheat germ, flax seeds, fennel, mint, caraway, anise, coriander, and mahaleb cherry (*Prunus mahaleb*), was checked using an electronic tongue (ET) sensor system (*Taha & Abu-Khalaf, 2020*). The results indicated that enough storage in the production facility was used since the homogeneity of the product was little impacted through storage duration by the ET signals.

Prediction of plant diseases is essential to reducing crop loss. Early disease prediction models were investigated for this purpose, where data on leaf wetness duration (LWD) was one of the critical components. Leaf wetness sensors were used to detect the length of time that leaves are moist (LWS) in tulsi (*Ocimum tenuiflorum*). Graphene oxide (GO) was employed as the sensor film to detect the water molecules on the leaf canopy, and the LWS was constructed on a flexible polyamide substrate. In tests of the manufactured GO LWS conducted in a lab setting, they exposed the whole sensing screen to water molecules and found that it responded to roughly 45,000% more than that of air (*Patle et al., 2022*).

## Variable rate technologies (VRT)

In precision agriculture, variable rate technologies (VRT) focuses on the mechanical use of materials for a particular location. Materials like fertilizers, seeds, chemicals, *etc.*, are precisely applied to a particular area and are used as a base of data collected by maps, GPS, and sensor systems. VRT can be map-based, *i.e.,* preplanned, cost-effective, non-destructive, and created using electromagnetic induction. In contrast, sensor-based VRT does not use maps but mounts crop sensors for decision-making that measure soil properties, fertilizer application, and other crop characteristics. Overall, VRT should be used to determine the optimum use of individual materials on particular lands to provide economic benefits to farmers by saving their fertilizers and chemicals, increasing production, reducing human error, and protecting the environment (*Clark & McGuckin, 1996*).

## Crop scouting

Commonly known as "field scouting", in which farmers take basic observations while traveling in crop fields. It is done at each lifespan stage to overcome the problems that affect crop yields, like weeds, pests, and disease infestations. Conventionally, crop scouting

was done by observing plants manually, keeping accounts using a field notebook, and using equipment like knives, lenses, and bags for sampling. Agriculture technology has advanced: farmers use smartphones instead of notebooks to keep accurate information on their fields. With the advancement of spatial technologies like GPS and UAV, the farmer did not need to go to the field and observe plants manually. These technologies provide accurate information (like where pests, disease, moisture, temperature, and poor soil are located) to the farmers that the naked eye cannot see and which parts of areas provide rapid assistance (*Hoxha, Bombaj & Ilbert, 2021*). Two main models of UAVs used in agriculture, *i.e.,* fixed-wing platforms controlled with a remote and GPS (*Garcia et al., 2013*), are very similar to planes and primarily used in large farm areas, while the multi-copter model is related to helicopters having multiple propellers (4–8) and is used in smaller farm areas (*Mustapa et al., 2014*). Infrared cameras in UAVs enhance information about farms, like the types of weeds & pests, soil moisture level, growth stage, plant health, *etc*. It helps farmers understand their crops not only field-by-field but also on a plant-by-plant basis (*Bansod et al., 2017*).

## Artificial intelligence and drone technology

Artificial intelligence (AI) technology is increasingly applied in the study of medicinal plants for tasks such as plant identification (*Ayumi et al., 2021*; *Kavitha et al., 2023*; *Roslan et al., 2023*), plant monitoring, climate suitability index (*Sadeghfam et al., 2024*), phytochemical analysis, disease diagnosis and drug discovery. It enables the identification of new therapeutic compounds and the analysis of extensive datasets regarding medicinal properties (*Khanna et al., 2024*), chemical composition, and growth patterns. This approach promotes sustainability and biodiversity by facilitating the discovery of novel compounds from underutilized or endangered species, optimizing cultivation practices, and ensuring product quality and safety. Ultimately, this helps conserve valuable plant resources and reduces reliance on synthetic chemicals. Medicinal plants are now being cultivated and monitored using drone technology. Advanced sensors and imaging technologies mounted on drones help in precision agriculture by monitoring plant health, detecting stress, and optimizing resource allocation. Multispectral and hyperspectral imaging capabilities can be used to evaluate growth parameters, identify diseases, and analyze the content of bioactive compounds in medicinal plants. This technology reduces labor costs, enhances productivity, and promotes sustainable farming practices, thereby preserving the quality of medicinal plants (*Ahmad et al., 2006*). The ability of drone technology to collect real-time, high-resolution data is revolutionizing medicinal plant resource production and management. Integrating AI and drone technology in medicinal plant cultivation is revolutionizing agriculture by improving productivity, sustainability, and quality. However, the sector faces challenges such as quality control, market access, data dependence, and the need for farmer training. By combining these advanced technologies with traditional practices, the medicinal plant industry can enhance sustainability and productivity while addressing the increasing consumer demand for natural products.

## CONCLUSIONS

Data gathered from research utilizing various manual sampling approaches and analytic techniques reveal a muddled picture, highlighting the necessity for the quick adoption of precision agriculture techniques. Through high-throughput phenotyping techniques, various technologies like remote sensing sensors (assessment of medicinal plant health), VRT, GPS, GIS, and crop scouting—made the tedious conventional phenotyping methods simple and effective. They support, enhance, and increase precise fertilizer applications, disease diagnosis, and water level prediction in the field by helping to deliver an effective and affordable solution. Utilizing these technologies will provide the researchers with the scientific tools to move from collecting wild aromatic and therapeutic plants with high ethnopharmacological interest to cultivating and restoring their habitats. This article discusses the many technologies (GPS, GIS, remote sensing, sensors system, *etc.*) employed in precision farming and how they might be applied to medicinal plant cultivation and manual harvesting.

### Funding
The authors received no funding for this work.

### Competing Interests
Mohammad Irfan is an Academic Editor for PeerJ.

### Author Contributions

- Vinay Kumar performed the experiments, prepared figures and/or tables, authored or reviewed drafts of the article, and approved the final draft.
- Ashwini Zadokar performed the experiments, prepared figures and/or tables, authored or reviewed drafts of the article, and approved the final draft.
- Pankaj Kumar conceived and designed the experiments, prepared figures and/or tables, authored or reviewed drafts of the article, and approved the final draft.
- Rohit Sharma performed the experiments, analyzed the data, prepared figures and/or tables, authored or reviewed drafts of the article, and approved the final draft.
- Rajnish Sharma analyzed the data, authored or reviewed drafts of the article, and approved the final draft.
- Mohammed Wasim Siddiqui analyzed the data, authored or reviewed drafts of the article, and approved the final draft.
- Mohammad Irfan conceived and designed the experiments, authored or reviewed drafts of the article, and approved the final draft.
- Rahul Chandora analyzed the data, authored or reviewed drafts of the article, and approved the final draft.

### Data Availability
This is a literature review.

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
