# Peer review of "Advancing medicinal plant agriculture: integrating technology and precision agriculture for sustainability"

_PeerJ, doi:10.7717/peerj.19058_

## Round 0.1 · original submission · Major Revisions

Please introduce the recommended changes suggested by the two reviewers, particularly in what concerns cutting-edge approaches linked to precision agriculture, like AI and drones. It is also relevant to highlight the current scenario of industrial application of medicinal plants.

Reviewer 1 ·

Basic reporting

- The manuscript is well-written, no English errors were noticed. No plagiarism was found but the journal can test to ensure the limited similarity with previous reports.
Only some comments are in the attached manuscript and here need to be addressed in order to improve the manuscript for publication:
- The concept were discussed in a professional way. the follow of the text was smooth.
- Citation in the text need to be re arranged and added close to the relevant information.
- In the tables most of literatures are old, the recent on was 2022. update of the literatures is needed up to 2025 you can find.
- The figures should be in high resolution and readable font.
- the Figure 2 need to provide examples on each technology used for specific purpose.

Experimental design

- The manuscript is well-written, no English errors were noticed. No plagiarism was found but the journal can test to ensure the limited similarity with previous reports.
Only some comments are in the attached manuscript and here need to be addressed in order to improve the manuscript for publication:
- The concept were discussed in a professional way. the follow of the text was smooth.
- Citation in the text need to be re arranged and added close to the relevant information.
- In the tables most of literatures are old, the recent on was 2022. update of the literatures is needed up to 2025 you can find.
- The figures should be in high resolution and readable font.
- the Figure 2 need to provide examples on each technology used for specific purpose.

Validity of the findings

- The review article is interesting and review the recent application of technology for medicinal plants. it will be useful for the readers in the filed.
However, the future step and suggested future research in this field should be add before the conclusion section.

Additional comments

no more comments

Annotated reviews are not available for download in order to protect the identity of reviewers who chose to remain anonymous.

Reviewer 2 ·

Basic reporting

1. To strengthen the agriculture sector, it is crucial to combine the eûorts of industrialization
(ûeld mechanization and fertilizer production), technology (genome editing and
manipulation), and the information sector (for the application of current technologies in
precision agriculture). However, the manuscript should added some new technology, such as AI, and drone technology, and also the current situation of the industry (Integrating
Technology and Precision Agriculture for Sustainability), the problems it faces, and how to develop it in the next step.
2. The types of market medicinal plants should be introduction in this manuscript.
3. These technology should be used in new medicinal plants found, discovery of new medicinal ingredients, utilization and protection of medicinal ingredients. And also the authors should add some examples of utilizing these technologies in these manuscript,.

Experimental design

Article content is within the Aims and Scope of the journal and article type.

Validity of the findings

Impact and novelty not assessed. Meaningful replication encouraged where rationale & benefit to literature is clearly stated.

Additional comments

1. To strengthen the agriculture sector, it is crucial to combine the eûorts of industrialization
(ûeld mechanization and fertilizer production), technology (genome editing and
manipulation), and the information sector (for the application of current technologies in
precision agriculture). However, the manuscript should added some new technology, such as AI, and drone technology, and also the current situation of the industry (Integrating
Technology and Precision Agriculture for Sustainability), the problems it faces, and how to develop it in the next step.
2. The types of market medicinal plants should be introduction in this manuscript.
3. These technology should be used in new medicinal plants found, discovery of new medicinal ingredients, utilization and protection of medicinal ingredients. And also the authors should add some examples of utilizing these technologies in these manuscript,.

---

## Round 0.2 · accepted · Accept

Thank you for keeping the quality standards of PeerJ.

Reviewer 1 ·

Basic reporting

The author responded to all comments, the manuscript can be accepted now if there is no negative reports from other reviewers

Experimental design

The author responded to all comments, the manuscript can be accepted now if there is no negative reports from other reviewers

Validity of the findings

The author responded to all comments, the manuscript can be accepted now if there is no negative reports from other reviewers

Additional comments

The author responded to all comments, the manuscript can be accepted now if there is no negative reports from other reviewers